# Parallel Sampling of Diffusion Models

**Andy Shih, Suneel Belkhale, Stefano Ermon, Dorsa Sadigh, Nima Anari**
Computer Science, Stanford University
{andyshih,belkhale,ermon,dorsa,anari}@cs.stanford.edu

## Abstract

Diffusion models are powerful generative models but suffer from slow sampling, often taking 1000 sequential denoising steps for one sample. As a result, considerable efforts have been directed toward reducing the number of denoising steps, but these methods hurt sample quality. Instead of reducing the number of denoising steps (trading quality for speed), in this paper we explore an orthogonal approach: can we run the denoising steps in parallel (trading compute for speed)? In spite of the sequential nature of the denoising steps, we show that surprisingly it is possible to parallelize sampling via Picard iterations, by guessing the solution of future denoising steps and iteratively refining until convergence. With this insight, we present ParaDiGMS, a novel method to accelerate the sampling of pretrained diffusion models by denoising multiple steps in parallel. ParaDiGMS is the first diffusion sampling method that enables trading compute for speed and is even compatible with existing fast sampling techniques such as DDIM and DPM-Solver. Using ParaDiGMS, we improve sampling speed by 2-4x across a range of robotics and image generation models, giving state-of-the-art sampling speeds of 0.2s on 100-step DiffusionPolicy and 14.6s on 1000-step StableDiffusion-v2 with no measurable degradation of task reward, FID score, or CLIP score.[1]

## 1 Introduction

Diffusion models [28, 9, 32] have demonstrated powerful modeling capabilities for image generation [34, 15, 24], molecular generation [38], robotic policies [12, 4], and other applications. The main limitation of diffusion models, however, is that sampling can be inconveniently slow. For example, the widely-used Denoising Diffusion Probabilistic Models (DDPMs) [9] can take 1000 denoising steps to generate one sample. In light of this, many works like DDIM [29] and DPMSolver [18] have proposed to improve sampling speed by reducing the number of denoising steps. Unfortunately, reducing the number of steps can come at the cost of sample quality.

We are interested in accelerating sampling of pretrained diffusion models without sacrificing sample quality. We ask the following question: rather than trading quality for speed, can we instead trade compute for speed? That is, could we leverage additional (parallel) compute to perform the same number of denoising steps faster? At first, this proposal seems unlikely to work, since denoising proceeds sequentially. Indeed, naïve parallelization can let us generate multiple samples at once (*improve throughput*), but generating a single sample with faster wall-clock time (*improving latency*) appears much more difficult.

We show that, surprisingly, it is possible to improve the sample latency of diffusion models by computing denoising steps in parallel. Our method builds on the idea of Picard iterations to guess the full denoising trajectory and iteratively refine until convergence. Empirically, we find that the number of iterations for convergence is much smaller than the number of steps. Therefore, by computing each iteration quickly via parallelization, we sample from the diffusion model much faster.

---

[1] Code for our paper can be found at `https://github.com/AndyShih12/paradigms`

37th Conference on Neural Information Processing Systems (NeurIPS 2023).

Our method **ParaDiGMS** (Parallel Diffusion Generative Model Sampling) is the first general method that allows for the tradeoff between compute and sampling speed of pretrained diffusion models. Remarkably, ParaDiGMS is compatible with classifier-free guidance [10] and with prior fast sampling methods [29, 18] that reduce the number of denoising steps. In other words, we present an orthogonal solution that can form combinations with prior methods (which we call ParaDDPM, ParaDDIM, ParaDPMSolver) to trade both compute and quality for speed.

We experiment with ParaDiGMS across a large range of robotics and image generation models, including Robosuite Square, PushT, Robosuite Kitchen, StableDiffusion-v2, and LSUN. Our method is strikingly consistent, providing an improvement across all tasks and all samplers (ParaDDPM, ParaDDIM, ParaDPMSolver) of around 2-4x speedup with no measurable decrease in quality on task reward, FID score, or CLIP score. For example, we improve the sample time of the 100-step action generation of DiffusionPolicy from 0.74s to 0.2s, and the 1000-step image generation of StableDiffusion-v2 on A100 GPUs from 50.0s to 14.6s. Our improvements also extend to few-step image generation, showing speedups for as low as 50-step DDIM. By enabling these faster sampling speeds without quality degradation, ParaDiGMS can enhance exciting applications of diffusion models such as real-time execution of diffusion policies or interactive generation of images.

## 2  Background

Diffusion models [28, 9] such as Denoising Diffusion Probabilistic Models (DDPM) were introduced as latent-variable models with a discrete-time forward diffusion process where $q(\boldsymbol{x}_0)$ is the data distribution, $\alpha$ is a scalar function, with latent variables $\{\boldsymbol{x}_t : t \leq T\}$ defined as

$$q(\boldsymbol{x}_t \mid \boldsymbol{x}_0) = \mathcal{N}(\boldsymbol{x}_t; \sqrt{\alpha(t)}\boldsymbol{x}_0, (1 - \alpha(t))\boldsymbol{I}).$$

By setting $\alpha(T)$ close to 0, $q(\boldsymbol{x}_T)$ converges to $\mathcal{N}(\boldsymbol{0}, \boldsymbol{I})$, allowing us to sample data $\boldsymbol{x}_0$ by using a standard Gaussian prior and a learned inference model $p_\theta(\boldsymbol{x}_{t-1} \mid \boldsymbol{x}_t)$. The inference model $p_\theta$ is parameterized as a Gaussian with predicted mean and time-dependent variance $\sigma_t^2$, and can be used to sample data by sequential denoising, i.e., $p_\theta(\boldsymbol{x}_0) = \prod_{t=1}^{T} p_\theta(\boldsymbol{x}_{t-1} \mid \boldsymbol{x}_t)$,

$$p_\theta(\boldsymbol{x}_{t-1} \mid \boldsymbol{x}_t) = \mathcal{N}\big(\boldsymbol{x}_{t-1}; \mu_\theta(\boldsymbol{x}_t), \sigma_t^2 \boldsymbol{I}\big). \tag{1}$$

Many works [32, 18] alternatively formulate diffusion models as a Stochastic Differential Equation (SDE) by writing the forward diffusion process in the form

$$\mathrm{d}\boldsymbol{x}_t = f(t)\boldsymbol{x}_t \mathrm{d}t + g(t)\mathrm{d}\boldsymbol{w}_t, \quad \boldsymbol{x}_0 \sim q(\boldsymbol{x}_0), \tag{2}$$

with the standard Wiener process $\boldsymbol{w}_t$, where $f$ and $g$ are position-independent functions that can be appropriately chosen to match the transition distribution $q(\boldsymbol{x}_t \mid \boldsymbol{x}_0)$ [18, 15]. These works use an important result from [2] that the reverse process of Eq. (2) takes on the form

$$\mathrm{d}\boldsymbol{x}_t = \underbrace{\big(f(t)\boldsymbol{x}_t - g^2(t)\nabla_{\boldsymbol{x}} \log q_t(\boldsymbol{x})\big)}_{\text{drift } s} \mathrm{d}t + \underbrace{g(t)}_{\sigma_t} \mathrm{d}\bar{\boldsymbol{w}}_t, \quad \boldsymbol{x}_T \sim q(\boldsymbol{x}_T), \tag{3}$$

where $\bar{\boldsymbol{w}}_t$ is the standard Wiener process in reverse time. This perspective allows us to treat the sampling process of DDPM as solving a discretization of the SDE where the DDPM inference model $p_\theta$ can be used to compute an approximation $p_\theta(\boldsymbol{x}_{t-1} \mid \boldsymbol{x}_t) - \boldsymbol{x}_t$ of the drift term in Eq. (3).

Since the focus of this paper is on sampling from a pretrained diffusion model, we can assume $p_\theta$ is given. For our purposes, we only need two observations about sampling from the reverse process in Eq. (3): we have access to an oracle that computes the drift at any given point, and the SDE has position-independent noise. We will use the latter observation in Section 3.

### 2.1  Reducing the number of denoising steps

DDPM typically uses a $T = 1000$ step discretization of the SDE. These denoising steps are computed sequentially and require a full pass through the neural network $p_\theta$ each step, so sampling can be extremely slow. As a result, popular works such as DDIM [29] and DPMSolver [18] have explored the possibility of reducing the number of denoising steps, which amounts to using a coarser discretization with the goal of trading sample quality for speed.

Empirically, directly reducing the number of steps of the stochastic sampling process of DDPM hurts sample quality significantly. Therefore many works [29, 32, 18] propose using an Ordinary Differential Equation (ODE) to make the sampling process more amenable to low-step methods. These works appeal to the *probability flow ODE* [20], a deterministic process with the property that the marginal distribution $p(\boldsymbol{x}_t)$ at each time $t$ matches that of the SDE, so in theory sampling from the probability flow ODE is equivalent to sampling from the SDE:

$$\mathrm{d}\boldsymbol{x}_t = \underbrace{\left( f(t)\boldsymbol{x}_t - \frac{1}{2}g^2(t)\nabla_{\boldsymbol{x}}\log q_t(\boldsymbol{x}) \right)}_{\text{drift } s} \mathrm{d}t, \quad \boldsymbol{x}_T \sim \mathcal{N}(\boldsymbol{0}, \boldsymbol{I}).$$

By sampling from the ODE instead of the SDE, works such as DDIM and DPMSolver (which have connections to numerical methods such as Euler and Heun) can reduce the quality degradation of few-step sampling (e.g., 50 steps).

As a summary, the current landscape of sampling from pretrained diffusion models is comprised of full-step DDPM or accelerated sampling techniques such as DDIM and DPMSolver that trade quality for speed by reducing the number of denoising steps.

**Notation**    We write $[a, b]$ to denote the set $\{a, a+1, \ldots, b\}$ and $[a, b)$ to denote the set $\{a, a+1, \ldots, b-1\}$ for $b > a$. We write $\boldsymbol{x}_{a:b}$ to denote the set $\{\boldsymbol{x}_i : i \in [a, b)\}$. Since our focus is on sampling, **in the rest of the paper we denote time as increasing for the reverse process.**

## 3    Parallel computation of denoising steps

Rather than investigating additional techniques for reducing the number of denoising steps, which can lead to quality degradation, we look towards other approaches for accelerating sampling. In particular, we investigate the idea of trading compute for speed: can we accelerate sampling by taking denoising steps in parallel? We clarify that our goal is not to improve sample *throughput* – that can be done with naïve parallelization, producing multiple samples at the same time. Our goal is to improve sample *latency* – minimize the wall-clock time required for generating a single sample by solving the denoising steps for a single sample in parallel. Lowering sample latency without sacrificing quality can greatly improve the experience of using diffusion models, and enable more interactive and real-time generation applications.

Parallelizing the denoising steps, however, seems challenging due to the sequential nature of existing sampling methods. The computation graph has a chain structure (Fig. 1), so it is difficult to propagate information quickly down the graph. To make headway, we present the method of Picard iteration, a technique for solving ODEs through fixed-point iteration. An ODE is defined by a drift function $s(\boldsymbol{x}, t)$ with position and time arguments, and initial value $\boldsymbol{x}_0$. In the integral form, the value at time $t$ can be written as

$$\boldsymbol{x}_t = \boldsymbol{x}_0 + \int_0^t s(\boldsymbol{x}_u, u)du.$$

In other words, the value at time $t$ must be the initial value plus the integral of the derivative along the path of the solution. This formula suggests a natural way of solving the ODE by starting with a guess of the solution $\{\boldsymbol{x}_t^k : 0 \le t \le 1\}$ at initial iteration $k = 0$, and iteratively refining by updating the value at every time $t$ until convergence:

$$\textbf{(Picard Iteration)} \qquad \boldsymbol{x}_t^{k+1} = \boldsymbol{x}_0^k + \int_0^t s(\boldsymbol{x}_u^k, u)du. \qquad (4)$$

Under mild conditions on $s$, such as continuity in time and Lipschitz continuity in position as in the well-known Picard-Lindelöf theorem, the iterates form a convergent sequence, and by the Banach fixed-point theorem, they converge to the unique solution of the ODE with initial value $\boldsymbol{x}_0$ [cf. 5]. To perform Picard iterations numerically, we can write the discretized form of Eq. (4) with step size $1/T$, for $t \in [0, T]$:

$$\boldsymbol{x}_t^{k+1} = \boldsymbol{x}_0^k + \frac{1}{T}\sum_{i=0}^{t-1} s(\boldsymbol{x}_i^k, i/T). \qquad (5)$$

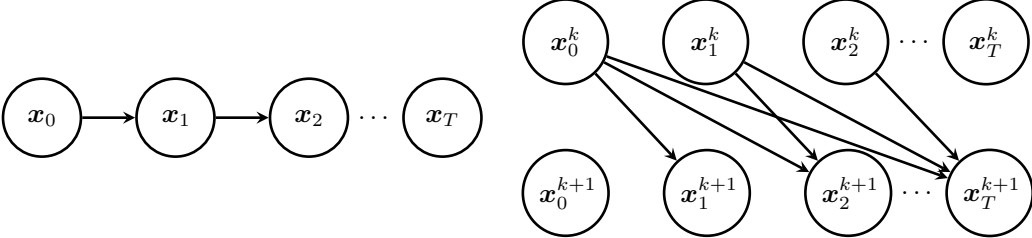

Figure 1: Computation graph of sequential sampling by evaluating $p_\theta(\boldsymbol{x}_{t+1} \mid \boldsymbol{x}_t)$, from the perspective of reverse time.

Figure 2: Computation graph of Picard iterations, which introduces skip dependencies.

Examining the iterative update rule in Eq. (5), we see that an update at time $t$ depends on all previous timesteps instead of just the previous timestep $t - 1$. This amounts to introducing skip dependencies in the computation graph (Fig. 2), which enables information to propagate quickly down the chain and accelerate sampling.

The key property of interest is that each Picard iteration can be parallelized by performing the expensive computations $\{s(\boldsymbol{x}_i^k, \frac{i}{T}) : i \in [0, T)\}$ in parallel and then, with negligible cost, collecting their outputs into prefix sums. Given enough parallel processing power, the sampling time then scales with the number of iterations $K$ until convergence, instead of the number of denoising steps $T$.

The number of iterations until convergence depends on the drift function $s$. More concretely, sequential evaluation can be written as a nested evaluation of functions $\boldsymbol{x}_{t+1}^\star = h_t(\ldots h_2(h_1(\boldsymbol{x}_0)))$ on the initial value $\boldsymbol{x}_0$ where $h_i(\boldsymbol{x}) = \boldsymbol{x} + s(\boldsymbol{x}, i/T)/T$. If, for all timesteps, the drift at the true solution can be accurately obtained using the drift at the current guess, then the parallel evaluation will converge in one step.

**Proposition 1.** *(Proof in Appendix A)*

$$s(\boldsymbol{x}_i^k, i/T) = s(h_{i-1}(\ldots h_2(h_1(\boldsymbol{x}_0))), i/T) \quad \forall i \le t \implies \boldsymbol{x}_{t+1}^{k+1} = \boldsymbol{x}_{t+1}^\star$$

It is also easy to see that even in the worst case, exact convergence happens in $K \le T$ iterations since the first $k$ points $\boldsymbol{x}_{0:k}$ must equal the sequential solution $\boldsymbol{x}_{0:k}^\star$ after $k$ iterations. In practice, the number of iterations until (approximate) convergence is typically much smaller than $T$, leading to a large empirical speedup.

The idea of Picard iterations is powerful because it enables the parallelization of denoising steps. Remarkably, Picard iterations are also fully compatible with prior methods for reducing the number of denoising steps. Recall that the drift term $s(\boldsymbol{x}_t, t/T)/T$ can be written as $h_t(\boldsymbol{x}_t) - \boldsymbol{x}_t$ and approximated using Euler discretization as $p_\theta(\boldsymbol{x}_{t+1} \mid \boldsymbol{x}_t) - \boldsymbol{x}_t$, but it can also be readily approximated using higher-order methods on $p_\theta$. In our experiments, we demonstrate the combination of the two axis of speedups to both reduce the number of denoising steps and compute the steps in parallel.

### 3.1 Practical considerations

Implementing Picard iteration on diffusion models presents a few practical challenges, the most important being that of GPU memory. Performing an iteration requires maintaining the entire array of points $\boldsymbol{x}_{0:T}$ over time, which can be prohibitively large to fit into GPU memory. To address this, we devise the technique of (mini-)batching which performs Picard iteration only on points $\boldsymbol{x}_{t:t+p}$ inside a window of size $p$ that can be chosen appropriately to satisfy memory constraints. Moreover, instead of iterating on $\boldsymbol{x}_{t:t+p}$ until convergence of the full window before advancing to the next window, we use a *sliding window* approach to aggressively shift the window forward in time as soon as the starting timesteps in the window converge.

One other issue is the problem of extending Picard iteration to SDEs, since we rely on the determinism of ODEs to converge to a fixed point. Fortunately, since the reverse SDE (Eq. (3)) has position-independent noise, we can sample the noise up-front and absorb these fixed noises into the drift of the (now deterministic) differential equation. Note that the resulting ODE is still Lipschitz continuous in position and continuous in time, guaranteeing the convergence of Picard iteration.

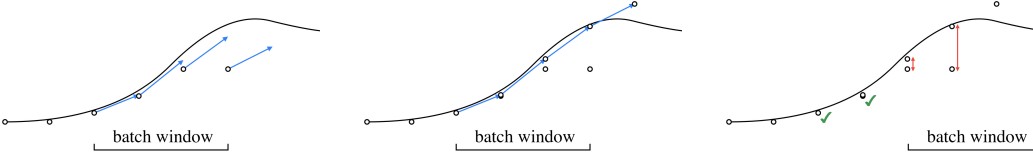

(a) Compute the drift of $\boldsymbol{x}_{t:t+p}^k$ on a batch window of size $p = 4$, in parallel

(b) Update the values to $\boldsymbol{x}_{t:t+p}^{k+1}$ using the cumulative drift of points in the window

(c) Determine how far to slide the window forward, based on the error $\|\boldsymbol{x}_i^{k+1} - \boldsymbol{x}_i^k\|^2$.

Figure 3: ParaDiGMS algorithm: accelerating an ODE solver by computing the drift at multiple timesteps in parallel. During iteration $k$, we process *in parallel* a batch window of size $p$ spanning timesteps $[t, t+p)$. The new values at a point $\boldsymbol{x}_{t+j}^{k+1}$ are updated based on the value $\boldsymbol{x}_j^k$ at the left end of the window plus the cumulative drift $1/T \sum_{i=t}^{t+j-1} s(\boldsymbol{x}_i^k, i/T)$ of points in the window. We then slide the window forward until the error is greater than our tolerance, and repeat for the next iteration.

Finally, we need to choose a stopping criterion for the fixed-point iteration, picking a low tolerance to avoid degradations of sample quality. A low enough tolerance ensures that the outcome of parallel sampling will be close to the outcome of the sequential sampling process in total variation distance.

**Proposition 2.** *(Proof in Appendix B) Assuming the iteration rule in Eq. (5) has a linear convergence rate with a factor $\geq 2$, using the tolerance $\|\boldsymbol{x}_t^K - \boldsymbol{x}_t^{K-1}\|^2 \leq 4\epsilon^2\sigma_t^2/T^2$ ensures that samples of $\boldsymbol{x}_T^K$ are drawn from a distribution with total variation distance at most $\epsilon$ from the DDPM model distribution of Eq. (1).*

The above is based on a worst-case analysis, and in our experiments, we find that using a much more relaxed tolerance such as[2] $\frac{1}{D}\|\boldsymbol{x}_i^{k+1} - \boldsymbol{x}_i^k\|^2 \leq \tau^2\sigma_i^2$, with $\tau = 0.1$ and $D$ being the dimensionality of data, gives reliable speedups without any measurable degradation in sample quality.

In Algorithm 1 we present the complete procedure of ParaDiGMS, incorporating sliding window over a batch, up-front sampling of noise, and tolerance of Picard iterations (Fig. 3). The loop starting on Line 4 performs a sliding window over the batch of timesteps $[t, t+p)$ in each iteration. Line 5 computes the drifts, which is the most compute-intensive part of the algorithm, but can be efficiently parallelized. Line 6 obtains their prefix sums in parallel to run the discretized Picard iteration update, and Lines 7-8 check the error values to determine how far forward we can shift the sliding window.

---

**Algorithm 1:** ParaDiGMS: parallel sampling via Picard iteration over a sliding window

**Input:** Diffusion model $p_\theta$ with variances $\sigma_t^2$, tolerance $\tau$, batch window size $p$, dimension $D$
**Output:** A sample from $p_\theta$

1  $t, k \leftarrow 0, 0$
2  $\boldsymbol{z}_i \sim \mathcal{N}(\boldsymbol{0}, \sigma_i^2\boldsymbol{I}) \quad \forall i \in [0, T)$              // Up-front sampling of noise (for SDE)
3  $\boldsymbol{x}_0^k \sim \mathcal{N}(\boldsymbol{0}, \boldsymbol{I}), \qquad \boldsymbol{x}_i^k \leftarrow \boldsymbol{x}_0^k \quad \forall i \in [1, p]$              // Sample initial condition from prior
4  **while** $t < T$ **do**
5  $\quad \boldsymbol{y}_{t+j} \leftarrow p_\theta(\boldsymbol{x}_{t+j}^k, t+j) - \boldsymbol{x}_{t+j}^k \quad \forall j \in [0, p)$              // Compute drifts in parallel
6  $\quad \boldsymbol{x}_{t+j+1}^{k+1} \leftarrow \boldsymbol{x}_t^k + \sum_{i=t}^{t+j} \boldsymbol{y}_i + \sum_{i=t}^{t+j} \boldsymbol{z}_i \quad \forall j \in [0, p)$              // Discretized Picard iteration
7  $\quad \text{error} \leftarrow \{\frac{1}{D}\|\boldsymbol{x}_{t+j}^{k+1} - \boldsymbol{x}_{t+j}^k\|^2 : \forall j \in [1, p)\}$              // Store error value for each timestep
8  $\quad \text{stride} \leftarrow \min\left(\{j : \text{error}_j > \tau^2\sigma_j^2\} \cup \{p\}\right)$              // Slide forward until we reach tolerance
9  $\quad \boldsymbol{x}_{t+p+j}^{k+1} \leftarrow \boldsymbol{x}_{t+p}^{k+1} \quad \forall j \in [1, \text{stride}]$              // Initialize new points that the window now covers
10 $\quad t \leftarrow t + \text{stride}, \qquad k \leftarrow k + 1$
11 $\quad p \leftarrow \min(p, T - t)$
12 **return** $\boldsymbol{x}_T^k$

---

The ParaDiGMS algorithm is directly compatible with existing fast sequential sampling techniques such as DDIM and DPMSolver, by swapping out the Euler discretization in Lines 5-6 for other

---

[2]For ODE methods (DDIM, DPMSolver) we still pick a tolerance value relative to the noise variance of the corresponding SDE of DDPM.

solvers, such as higher-order methods like Heun. As we see in our experiments, the combination of reducing the number of steps and solving the steps in parallel leads to even faster sample generation.

## 4 Experiments

We experiment with our method ParaDiGMS on a suite of robotic control tasks [4] including Square [41], PushT, Franka Kitchen [7], and high-dimensional image generation models including StableDiffusion-v2 [24] and LSUN Church and Bedroom [39]. We observe a consistent improvement of around 2-4x speedup relative to the sequential baselines without measurable degradation in sample quality as measured by task reward, FID score, or CLIP score.

### 4.1 Diffusion policy

Recently, a number of works have demonstrated the advantages of using diffusion models in robotic control tasks for flexible behavior synthesis or robust imitation learning on multimodal behavior [12, 4, 22, 36]. We follow the setup of DiffusionPolicy [4], which is an imitation learning framework that models action sequences. More specifically, DiffusionPolicy first specifies a prediction horizon $h$ and a replanning horizon $r$. At each environment step $l$, DiffusionPolicy conditions on a history of observations and predicts a sequence of actions $\{a_{l:l+h}\}$. Then, the policy executes the first $r$ actions $\{a_{l:l+r}\}$ of the prediction. Therefore, for an episode of length $L$ and scheduler with $T$ steps, executing a full trajectory can take $T \times L/r$ denoising steps over a dimension of $|a| \times h$, which can be inconveniently slow.

We examine our method on the Robosuite Square, PushT, and Robosuite Kitchen tasks. Each environment uses a prediction horizon of 16, and replanning horizon 8. The Square task uses state-based observations with a maximum trajectory length of 400 and a position-based action space of dimensionality 7. This means the diffusion policy takes 50 samples per episode, with each sample being a series of denoising steps over a joint action sequence of dimension 112. The PushT task also uses state-based observations and has a maximum trajectory length of 300 and action space of 2, which results in 38 samples with denoising steps over a joint action sequence of dimension 32. Lastly, the Kitchen task uses vision-based observations and has a maximum trajectory length of 1200 with an action space of 7, giving 150 samples per episode and denoising steps over a joint action sequence of dimensionality 112. For all three tasks, we use a convolution-based architecture.

The DDPM scheduler in DiffusionPolicy [4] uses 100 step discretization, and the DDIM/DPMSolver schedulers use 15 step discretization. For example, a trajectory in the Kitchen task requires $1200/8 = 150$ samples, which amounts to $150 \times 100 = 15000$ denoising steps over an action sequence of dimensionality 112 with the DDPM scheduler.

In Table 1, we present results on DDPM, DDIM, DPMSolver, and their parallel variants (ParaDDPM, ParaDDIM, ParaDPMSolver) when combined with ParaDiGMS. We plot the model evaluations (number of calls to the diffusion model $p_\theta$), the task reward, and the sampling speed reported in time per episode. Although parallelization increases the total number of necessary model evaluations, the sampling speed is more closely tied to the number of parallel iterations, which is much lower. We see that ParaDDPM gives a speedup of 3.7x, ParaDDIM gives a speedup of 1.6x, and ParaDPMSolver gives a speedup of 1.8x, without a decrease in task reward. Table 2 presents similar findings on the PushT task, where we see speedups on all three methods with up to 3.9x speedup on ParaDDPM.

Table 1: Robosuite Square with ParaDiGMS using a tolerance of $\tau = 0.1$ and a batch window size of 20 on a single A40 GPU. Reward is computed using an average of 200 evaluation episodes, with sampling speed measured as time to generate $400/8 = 50$ samples.

| Robosuite Square | Sequential | | | ParaDiGMS | | | | Speedup |
| | Model Evals | Reward | Time per Episode | Model Evals | Parallel Iters | Reward | Time per Episode | |
| --- | --- | --- | --- | --- | --- | --- | --- | --- |
| DDPM | 100 | $0.85 \pm 0.03$ | 37.0s | 392 | 25 | $0.85 \pm 0.03$ | 10.0s | 3.7x |
| DDIM | 15 | $0.83 \pm 0.03$ | 5.72s | 47 | 7 | $0.85 \pm 0.03$ | 3.58s | 1.6x |
| DPMSolver | 15 | $0.85 \pm 0.03$ | 5.80s | 41 | 6 | $0.83 \pm 0.03$ | 3.28s | 1.8x |

Table 2: PushT task with ParaDiGMS using a tolerance of $\tau = 0.1$ and a batch window size of 20 on a single A40 GPU. Reward is computed using an average of 200 evaluation episodes, with sampling speed measured as time to generate $\lceil 300/8 \rceil = 38$ samples.

| PushT | Sequential | | | ParaDiGMS | | | | |
|---|---|---|---|---|---|---|---|---|
| | Model Evals | Reward | Time per Episode | Model Evals | Parallel Iters | Reward | Time per Episode | Speedup |
| DDPM | 100 | $0.81 \pm 0.03$ | 32.3s | 386 | 24 | $0.83 \pm 0.03$ | 8.33s | 3.9x |
| DDIM | 15 | $0.78 \pm 0.03$ | 4.22s | 46 | 7 | $0.77 \pm 0.03$ | 2.54s | 1.7x |
| DPMSolver | 15 | $0.79 \pm 0.03$ | 4.22s | 40 | 6 | $0.79 \pm 0.03$ | 2.15s | 2.0x |

Table 3: FrankaKitchen with ParaDiGMS using a tolerance of $\tau = 0.1$ and a batch window size of 20 on a single A40 GPU. Reward is computed using an average of 200 evaluation episodes, with sampling speed measured as time to generate $1200/8 = 150$ samples.

| Franka Kitchen | Sequential | | | ParaDiGMS | | | | |
|---|---|---|---|---|---|---|---|---|
| | Model Evals | Reward | Time per Episode | Model Evals | Parallel Iters | Reward | Time per Episode | Speedup |
| DDPM | 100 | $0.85 \pm 0.03$ | 112s | 390 | 25 | $0.84 \pm 0.03$ | 33.3s | 3.4x |
| DDIM | 15 | $0.80 \pm 0.03$ | 16.9s | 47 | 7 | $0.80 \pm 0.03$ | 9.45s | 1.8x |
| DPMSolver | 15 | $0.79 \pm 0.03$ | 17.4s | 41 | 6 | $0.80 \pm 0.03$ | 8.89s | 2.0x |

The final robotics task we study is FrankaKitchen, a harder task with predicted action sequences of dimension 112 and an episode length of 1200. In Table 3 we notice some decline in performance when sampling with a reduced number of steps using DDIM and DPMSolver. On the other hand, ParaDDPM is able to maintain a high task reward. Similar to before, ParaDiGMS consistently achieves a speedup across all 3 sampling methods, giving a speedup of 3.4x with ParaDDPM, 1.8x with ParaDDIM, and 2.0x with ParaDPMSolver. These improvements translate to a significant decrease in the time it takes to roll out an episode in the Kitchen task from 112s to 33.3s.

## 4.2 Diffusion image generation

Next, we apply parallel sampling to diffusion-based image generation models, both for latent-space and pixel-space models. For latent-space models, we test out StableDiffusion-v2[3] [26, 24], which generates 768x768 images using a diffusion model on a 4x96x96 latent space. For pixel-space models, we study pretrained models on LSUN Church[4]/Bedroom[5] from Huggingface [9, 35], which run a diffusion model directly over the 3x256x256 pixel space.

### 4.2.1 Latent-space diffusion models

Even with the larger image models, there is no issue fitting a batch window size of 20 on a single GPU for parallelization. However, the larger model requires more compute bandwidth, so the parallel efficiency quickly plateaus as the batch window size increases, as the single GPU becomes bottlenecked by floating-point operations per second (FLOPS). Therefore, for image models, we leverage multiple GPUs to increase FLOPS and improve the wall-clock sampling speed.

In Fig. 4 we examine the net speedup of ParaDDPM relative to DDPM on StableDiffusion-v2 using 1000-step diffusion on A100 GPUs. The net speedup is determined by the interplay between *algorithm inefficiency* and *hardware efficiency*. Algorithm inefficiency refers to the relative number of model evaluations of ParaDDPM compared to DDPM, which arises from the parallel algorithm taking multiple iterations until convergence. We see in Fig. 4e that as the batch window size grows, ParaDDPM can require 2-3x more model evaluations. On the other hand, hardware efficiency refers to the relative empirical speedup of performing a batch of model evaluations. For example, in Fig. 4e we see that evaluating a batch window size of 80 on 4 GPUs (20 per GPU) is roughly 5x faster than performing 80 model evaluations sequentially. In Fig. 4f, we divide the hardware efficiency by the

---

[3] https://huggingface.co/stabilityai/stable-diffusion-2
[4] https://huggingface.co/google/ddpm-ema-church-256
[5] https://huggingface.co/google/ddpm-ema-bedroom-256

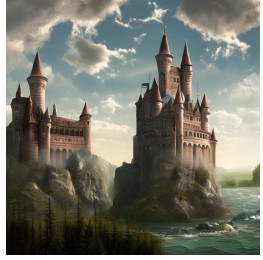 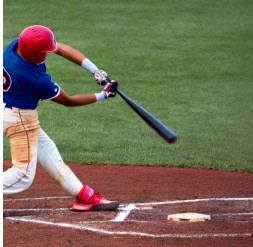 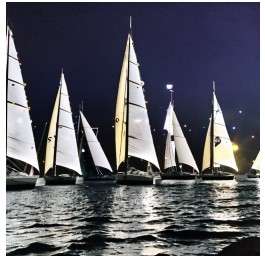 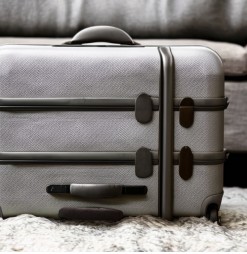

(a) "a beautiful castle, matte painting"

(b) "a batter swings at a pitch during a baseball game"

(c) "several sail boats in the water at night"

(d) "a grey suitcase sits in front of a couch"

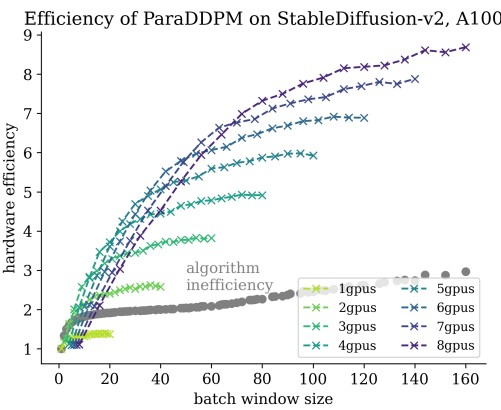 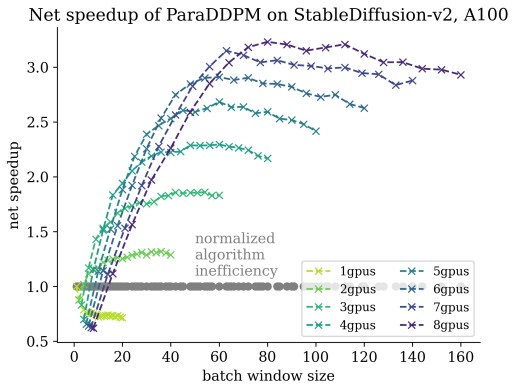

(e) Hardware efficiency overtakes algorithm inefficiency as number of GPUs increase.

(f) Over 3x net wall-clock speedup for 1000-step ParaD-DPM

Figure 4: StableDiffusion-v2 generating text-conditioned 768x768 images by running ParaDDPM over a 4x96x96 latent space for 1000 steps, on A100 GPUs. In Fig. 4e algorithm inefficiency in gray denotes the relative number of model evaluations required as the parallel batch window size increases. The colored lines denote the hardware efficiency provided by the multi-GPUs. As the batch window size increases, the hardware efficiency overtakes the algorithm inefficiency. In Fig. 4f we normalize the algorithm inefficiency to 1, to show the net wall-clock speedup of parallel sampling.

algorithm inefficiency to obtain the net relative speedup of ParaDDPM over DDPM. We observe over 3x speedup by using a batch window size of 80 spread across 8 A100s. Finally, in Table 4 we verify that ParaDiGMS increases sampling speed for ParaDDPM, ParaDDIM, and ParaDPMSolver without degradation in sample quality as measured by CLIP score [8] on ViT-g-14 [23, 11].

One important consideration is that the algorithm inefficiency is agnostic to the choice of GPU. Therefore, as the parallel efficiency of GPUs in the future improve for large batch window sizes, we will see an even larger gap between hardware efficiency and algorithm inefficiency. With enough hardware efficiency, the wall-clock time of sampling will be limited only by the number of parallel iterations, leading to much larger net speedup. For example, observe that in Table 4 the number of parallel iterations of ParaDDPM is 20x smaller than the number of sequential steps.

### 4.2.2 Pixel-space diffusion models

Next, we test out ParaDiGMS on pretrained diffusion models on LSUN Church and Bedroom, which perform diffusion directly on a 3x256x256 pixel space. In Fig. 5 in Appendix D, we plot the net speedup of 1000-step ParaDDPM by dividing the hardware efficiency by the algorithm inefficiency. We observe a similar trend of over 3x speedup when using multiple GPUs. Finally, we verify in Table 5 that ParaDiGMS maintains the same sample quality as the baseline methods as measured by FID score on 5000 samples of LSUN Church[6].

---

[6]DPMSolver is not yet integrated with the LSUN model in the Diffusers library, so we omit its comparison.

Table 4: Evaluating CLIP score of ParaDiGMS on StableDiffusion-v2 over 1000 random samples from the COCO2017 captions dataset, with classifier guidance $w = 7.5$. CLIP score is evaluated on ViT-g-14, and sample speed is computed on A100 GPUs.

| Stable Diffusion-v2 | Sequential | | | ParaDiGMS | | | | | |
| | Model Evals | CLIP Score | Time per Sample | Tol. $\tau$ | Model Evals | Parallel Iters | CLIP Score | Time per Sample | Speedup |
|---|---|---|---|---|---|---|---|---|---|
| DDPM | 1000 | 32.1 | 50.0s | 1e-1 | 2504 | 50 | 32.1 | 14.6s | 3.4x |
| DPMSolver | 200 | 31.7 | 10.3s | 1e-1 | 422 | 15 | 31.7 | 2.6s | 4.0x |
| DDIM | 200 | 31.9 | 10.3s | 1e-1 | 432 | 15 | 31.9 | 2.6s | 4.0x |
| DDIM | 100 | 31.9 | 5.3s | 5e-2 | 229 | 19 | 31.9 | 2.0s | 2.7x |
| DDIM | 50 | 31.9 | 2.6s | 5e-2 | 91 | 17 | 31.9 | 1.1s | 2.4x |
| DDIM | 25 | 31.7 | 1.3s | 1e-2 | 93 | 17 | 31.7 | 1.0s | 1.3x |

We highlight that 500-step DDIM gives a noticeably worse FID score than 1000-step DDPM, whereas using ParaDDPM allows us to maintain the same sample quality as DDPM while accelerating sampling (to be two times faster than 500-step DDIM). The ability to generate an image without quality degradation in 6.1s as opposed to 24.0s can significantly increase the viability of interactive image generation for many applications.

Table 5: Evaluating FID score (lower is better) of ParaDiGMS on LSUN Church using 5000 samples. Sample speed is computed on A100 GPUs. We use tolerance 5e-1 for DDPM and 1e-3 for DDIM.

| LSUN Church | Sequential | | | ParaDiGMS | | | | |
| | Model Evals | FID Score | Time per Sample | Model Evals | Parallel Iters | FID Score | Time per Sample | Speedup |
|---|---|---|---|---|---|---|---|---|
| DDPM | 1000 | 12.8 | 24.0s | 2583 | 45 | 12.9 | 6.1s | 3.9x |
| DDIM | 500 | 15.5 | 12.2s | 1375 | 41 | 15.3 | 3.7s | 3.3x |
| DDIM | 100 | 15.1 | 2.5s | 373 | 23 | 14.8 | 1.0s | 2.5x |
| DDIM | 50 | 15.3 | 1.2s | 168 | 17 | 15.7 | 0.7s | 1.7x |
| DDIM | 25 | 15.6 | 0.6s | 70 | 15 | 15.9 | 0.6s | 1.0x |

## 4.3 Related work

Apart from DDIM [29] and DPMSolver [18], there are a number of other fast sampling techniques for pretrained diffusion models [14, 17, 19, 40]. Most of these techniques are based on higher-order ODE-solving and should also be compatible with parallelization using ParaDiGMS. Other lines of work focus on distilling a few-step model [25, 21, 30] or learning a sampler [37], but these methods are more restrictive as they require additional training.

Besides diffusion models, many works have studied accelerating the sampling of autoregressive models using various approaches such as parallelization [33, 31], distillation [13], quantization [6], or rejection sampling [3, 16]. Of particular note is [31], which samples from autoregressive models by iterating on a system of equations until convergence. While these methods are not immediately applicable to diffusion models due to the differences in computational structure and inference efficiency of autoregressive models, there may be potential for further investigation.

Parallelization techniques similar to Picard iteration have been explored in theoretical works for sampling from log-concave [27] and determinantal distributions [1]. Our work is the first application of parallel sampling on diffusion models, enabling a new axis of trading compute for speed.

## 5 Conclusion

**Limitations** Since our parallelization procedure requires iterating until convergence, the total number of model evaluations increases relative to sequential samplers. Therefore, our method is not suitable for users with limited compute who wish to maximize sample throughput. Nevertheless, sample latency is often the more important metric. Trading compute for speed with ParaDiGMS

makes sense for many practical applications such as generating images interactively, executing robotic policies in real-time, or serving users who are insensitive to the cost of compute.

Our method is also an approximation to the sequential samplers since we iterate until the errors fall below some tolerance. However, we find that using ParaDiGMS with the reported tolerances results in no measurable degradations of sample quality in practice across a range of tasks and metrics. In fact, on more difficult metrics such as FID score on LSUN Church, ParaDDPM gives both higher sample quality and faster sampling speed than 500-step DDIM.

**Discussion**    We present ParaDiGMS, the first accelerated sampling technique for diffusion models that enables the trade of compute for speed. ParaDiGMS improves sampling speed by using the method of Picard iterations, which computes multiple denoising steps in parallel through iterative refinement. Remarkably, ParaDiGMS is compatible with existing sequential sampling techniques like DDIM and DPMSolver, opening up an orthogonal axis for optimizing the sampling speed of diffusion models. Our experiments demonstrate that ParaDiGMS gives around 2-4x speedup over existing sampling methods across a range of robotics and image generation models, without sacrificing sample quality. As GPUs improve, the relative speedup of ParaDiGMS will also increase, paving an exciting avenue of trading compute for speed that will enhance diffusion models for many applications.

# 6    Acknowledgments

This research was supported in part by NSF (#1651565, #2045354, #2125511), ARO (W911NF-21-1-0125), ONR (N00014-23-1-2159, N00014-22-1-2293), CZ Biohub, HAI.

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

## A Proof of exact convergence

*Proof of Proposition 1.* Assume by induction that $\boldsymbol{x}_t^{k+1} = \boldsymbol{x}_t^\star$. Then

$$
\begin{aligned}
\boldsymbol{x}_{t+1}^{k+1} &= \boldsymbol{x}_0^k + \frac{1}{T}\sum_{i=0}^{t} s(\boldsymbol{x}_i^k, \frac{i}{T}) \\
&= \left( \boldsymbol{x}_0^k + \frac{1}{T}\sum_{i=0}^{t-1} s(\boldsymbol{x}_i^k, \frac{i}{T}) \right) + \frac{1}{T} s(\boldsymbol{x}_t^k, \frac{t}{T}) \\
&= \boldsymbol{x}_t^{k+1} + \frac{1}{T} s(\boldsymbol{x}_t^k, \frac{t}{T}) \\
&= \boldsymbol{x}_t^{k+1} + \frac{1}{T} s(h_{t-1}(\dots h_2(h_1(\boldsymbol{x}_0))), \frac{t}{T}) \\
&= \boldsymbol{x}_t^\star + \frac{1}{T} s(\boldsymbol{x}_t^\star, \frac{t}{T}) = \boldsymbol{x}_{t+1}^\star. \qquad \square
\end{aligned}
$$

## B Total variation analysis

*Proof of Proposition 2.* A linear convergence rate with factor $\geq 2$ ensures our error from the solution $\boldsymbol{x}_t^\star$ given by sequential sampling at each timestep $t$ is bounded by the chosen tolerance:

$$
\|\boldsymbol{x}_t^K - \boldsymbol{x}_t^\star\|^2 \leq \lim_{n\to\infty} \sum_{j=K+1}^{n} \|\boldsymbol{x}_t^j - \boldsymbol{x}_t^{j-1}\|^2 \leq \lim_{n\to\infty} \sum_{j=K+1}^{n} \frac{1}{2^{j-K}} \|\boldsymbol{x}_t^K - \boldsymbol{x}_t^{K-1}\|^2 \leq \|\boldsymbol{x}_t^K - \boldsymbol{x}_t^{K-1}\|^2.
$$

Then, for each timestep $t$, since the inference model samples from a Gaussian with variance $\sigma_t^2$, we can bound the total variation distance:

$$
\begin{aligned}
d_{\mathrm{TV}}(\mathcal{N}(\boldsymbol{x}_t^K, \sigma_t^2 \boldsymbol{I}), \mathcal{N}(\boldsymbol{x}_t^\star, \sigma_t^2 \boldsymbol{I})) &\leq \sqrt{\frac{1}{2} \mathcal{D}_{\mathrm{KL}}(\mathcal{N}(\boldsymbol{x}_t^K, \sigma_t^2 \boldsymbol{I}) \,\|\, \mathcal{N}(\boldsymbol{x}_t^\star, \sigma_t^2 \boldsymbol{I}))} \\
&= \sqrt{\frac{\|\boldsymbol{x}_t^K - \boldsymbol{x}_t^\star\|^2}{4\sigma_t^2}} \leq \sqrt{\frac{\|\boldsymbol{x}_t^K - \boldsymbol{x}_t^{K-1}\|^2}{4\sigma_t^2}} \leq \frac{\epsilon}{T}.
\end{aligned}
$$

Finally, we make use of the data processing inequality, that $d_{\mathrm{TV}}(f(P), f(Q)) \leq d_{\mathrm{TV}}(P, Q)$, so the total variation distance $d_t$ between the sample and model distribution after $t$ timesteps does not increase when transformed by $p_\theta$. Then by the triangle inequality, we get that $d_t \leq d_{t-1} + \epsilon/T$. giving a total variation distance $d_T$ of at most $T\epsilon/T = \epsilon$ for the final timestep $T$. $\qquad \square$

## C Parallel Computation on Multiple GPUs

We experimented with two different approaches to implementing parallel computation on multiple GPUs in PyTorch: 1) using `torch.nn.DataParallel` and 2) using `torch.multiprocessing`. DataParallel is very easy to implement, but performed more poorly than multiprocessing. For multiprocessing, we use a producer/consumer design where we spawn a single producer process on GPU 0 to run the main loop in Algorithm 1, and $N$ consumer processes (one for each of the $N$ GPUs) to run Line 5 of Algorithm 1 in parallel. Empirically, this worked better than dedicating the producer process with its own GPU and using only $N-1$ consumer processes, since the producer process requires very little GPU resources.

# D   Additional experiments

## D.1   LSUN Church and LSUN Bedroom

In Fig. 5 we plot wallclock speedup of ParaDDPM as a function of batch window size and number of GPUs on LSUN Church and LSUN Bedroom. These experiments on LSUN show a similar trend as the experiments on StableDiffusion-v2 in Fig. 4f.

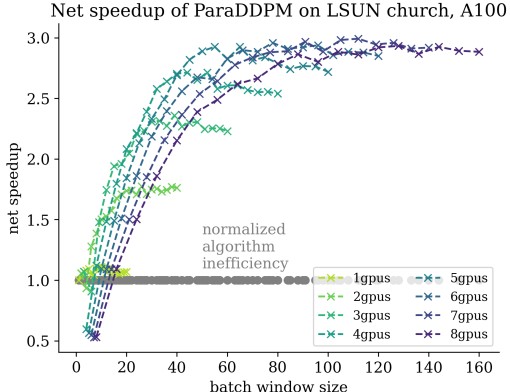
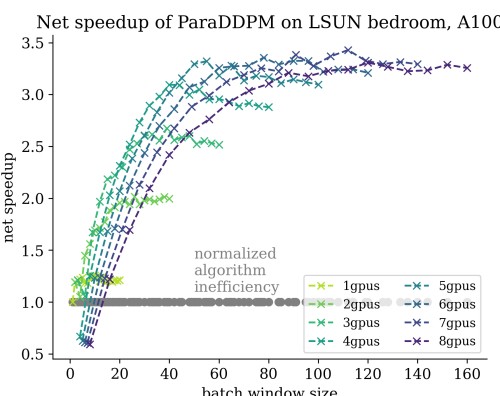

(a) LSUN Church, close to 3x net wall clock speedup with 1000-step ParaDDPM using DataParallel

(b) LSUN Bedroom, over 3x net wall clock speedup with 1000-step ParaDDPM using DataParallel

Figure 5: Unconditional generation of 256x256 images on diffusion models pretrained on the LSUN Church and Bedroom dataset, running ParaDDPM for 1000 steps on A100 GPUs. We plot the net speedup after dividing the hardware efficiency by the algorithm inefficiency as the batch window size increases. Note that Table 5 shows better speedups because for Table 5 we use a better parallel implementation with multiprocessing instead of DataParallel.

## D.2   Ablation

In Table 6, we run an ablation study on the tolerance parameter $\tau$ for 200-step ParaDDIM. A lower tolerance means the algorithm takes more parallel iterations and attains better sample quality, whereas a higher tolerance means the algorithm slides the batch window forward more aggressively leading to fewer iterations and faster sampling. We see in Table 6 that for 200-step ParaDDIM on StableDiffusion-v2, a good choice of $\tau$ is 1e-1. Lower tolerance levels give less speedup without noticeable increase in CLIP score, and higher tolerance levels exhibit a drop in CLIP score.

Table 6: Ablation on the effect of error tolerance on sample quality and speed on StableDiffusion-v2. Samples are generated using ParaDiGMS with 200-step DDIM. CLIP score is computed over 1000 samples. Sample speed is computed on A100 GPUs.

| StableDiffusion-v2 | Steps | Tolerance $\tau$ | Parallel Iters | CLIP Score | Time per Sample | Speedup |
|---|---|---|---|---|---|---|
| DDIM | 200 | sequential | 200 | 31.9 | 10.3s | - |
| DDIM | 200 | 5e-3 | 36 | 31.9 | 7.4s | 1.4x |
| DDIM | 200 | 1e-2 | 28 | 31.9 | 5.1s | 2.0x |
| DDIM | 200 | 5e-2 | 19 | 31.9 | 3.3s | 3.1x |
| DDIM | 200 | 1e-1 | 15 | 31.9 | 2.6s | 4.0x |
| DDIM | 200 | 5e-1 | 13 | 31.5 | 2.1s | 4.9x |
| DDIM | 200 | 1e-0 | 12 | 31.5 | 1.9s | 5.4x |