# OpenReview forum: "Parallel Sampling of Diffusion Models"
_NeurIPS.cc/2023/Conference — NeurIPS 2023 spotlight_

### Official Review · Reviewer_psnQ · 2023-07-04

**Soundness:** 4 excellent
**Presentation:** 4 excellent
**Contribution:** 4 excellent
**Rating:** 8
**Confidence:** 4

**Summary:**

The paper introduces a new method for speeding up (reduced latency) sampling of diffusion models by sampling all time steps in parallel from some initialization and then iterating this procedure until convergence (Picard Iteration). Modifications are made for implementation efficiency such as a sliding window. The method is evaluated on control tasks and large scale image datasets with a consistent reduction in latency at the expense of increased compute being reported.

**Strengths:**

This paper is very good. The problem being tackled is highly significant and I think the approach will be widely used to easily achieve a reduction in latency when extra compute is available. The method is clearly presented with a nice bit of theoretical analysis. The experiments on practically releveant datasets clearly show the benefits of the method. I highly encourage acceptance.

**Weaknesses:**

I think it would have been nice to pick a suitably small problem and fully explore the compute/latency tradeoff by varying the window size. I appreciate the current experiment in Figure 4 which does nicely show this trade-off in this compute intensive regime however the picture is slightly obscured by the necessity of also considering GPU flop/s saturation (which is an important consideration in its own right I agree). I think a simpler experiment where you can vary the window size all the way up to the number of denoising steps would benefit the paper where we can see a potential 20x reduction in latency if DDPM is integrated with 1000 steps and the Picard iterations converge in around 50 steps.

For the experiments on StableDiffusion, it is not clear to what extent the method's benefits extend into the very low model evaluation regime. DPMSolver should be able to go to very few model evaluations and retain decent quality, in this regime, how does your method compare? It is ok if the model speedup is less in this case, but it would be good to know the extent to which the method can provide benefits.

I think it would be good to stick with the terminology batch window size as you go through the experiments. When getting to section 4.2.1 it is a little confusing to start talking about batch sizes which may imply that you are generating entirely independent trajectories in parallel but I assume you still mean the batch containing the points in the window.

Edit after rebuttal: I have read the author's response and my concerns have been addressed. I will keep my score of 8.

**Questions:**

In proposition 2, how realistic is the assumption of the convergence rate of the Picard iterations in the context of integrating the probability flow ODE from a trained diffusion model?

This has been investigated somewhat between the two experiments but what would you say is the interaction between the number of Picard iterations and the number of steps used to integrate the ODE. Do you believe there could be any relation where more steps could be helpful to reduce the number of total Picard iterations or are these completely orthogonal quantities?

**Limitations:**

The method is obviously more expensive than other diffusion models due to all steps being taken in parallel but this trade-off is well explained. Further, as explained above, the limits of when the method does not actually provide a speedup could have been better investigated.

---

> ### Author Rebuttal · Authors · 2023-08-09
>
> We thank the reviewer for the review.
>
> &nbsp;
>
> > **pick a suitably small problem and fully explore the compute/latency tradeoff by varying the window size… vary the window size all the way up to the number of denoising steps.**
>
> Thank you for the suggestion. To better explore this tradeoff of varying window sizes, we use the Square task from Table 1 using a single GPU. This task uses a smaller diffusion model, so the effects of FLOPS saturation is less noticeable (but unfortunately still present for large window sizes). We see that the relative speedup increases up until a batch window size of 150, and then decreases as FLOPS saturation kicks in.
>
> Square Task
> | batch window size | samples/sec | relative speedup | Parallel Iters |
> |-------------------|-------------|------------------|------------|
> | 1 (sequential) | 1.09              | 1x          | 1000             |
> | 50             | 9.07              | 8.3x        | 76               |
> | 100            | 11.11             | 10.2x       | 56               |
> | 150            | 12.02             | 11.0x       | 49               |
> | 200            | 11.84             | 10.9x       | 47               |
> | 300            | 11.40             | 10.4x       | 44               |
> | 500            | 9.64              | 8.9x        | 43               |
> | 1000           | 7.69              | 7.0x        | 43               |
>
>
> For batch window size of 150, we see that (sequential steps) / (parallel iterations) = 1000 / 49, giving an over 20x theoretical speedup, which is similar to that of image models as well (L252 in the paper). Unlike image models, however, FLOPS saturation is less of an issue here, so we are able to get a 11x speedup when using a batch window size of 150. This highlights the fact that the relative speedup of ParaDiGMS can increase as the effect of FLOPS saturation diminishes with better hardware in the future.
>
> &nbsp;
>
> > **to what extent the method's benefits extend into the very low model evaluation regime? in this regime, how does your method compare?**
>
> Please see the shared response on “Speedup with few steps”. We present exciting developments where we implement custom multiprocessing for ParaDiGMS and are now able to see a 2.5x speedup on the most widely used setting of 50-step DDIM, and a 1.3x speedup even for 25-step DDIM.
>
> &nbsp;
>
> > **the limits of when the method does not actually provide a speedup could have been better investigated**
>
> Based on the new experiments, we see a drop in speedup for 25-step DDIM (1.3x) when compared to the speedup for 50-step DDIM (2.5x). This suggests that 25-step DDIM is roughly the limits of the current method on large models such as stable diffusion. On smaller models such as diffusion policy, we achieve large speedups even for 15-step DDIM.
>
> &nbsp;
>
> > **stick with the terminology batch window size as you go through the experiments. When getting to section 4.2.1 it is a little confusing to start talking about batch sizes**
>
> Thanks for the catch, we will be consistent in referring to it as the batch window size.
>
> &nbsp;
>
> > **In proposition 2, how realistic is the assumption of the convergence rate of the Picard iterations in the context of integrating the probability flow ODE from a trained diffusion model?**
>
> We believe that this assumption is very realistic in the setting of diffusion models, and empirically seems to be satisfied in the experiments we tried. One way to see this is that the tolerance used in our experiments to attain equal sample quality is much looser than the necessary tolerance stated in proposition 2. This suggests that the convergence rate in practice is much faster than the convergence rate stated in the assumption.
>
> &nbsp;
>
> > **what would you say is the interaction between the number of Picard iterations and the number of steps used to integrate the ODE. Do you believe there could be any relation where more steps could be helpful to reduce the number of total Picard iterations or are these completely orthogonal quantities?**
>
> Based on our exploration, empirically more steps for integrating the ODE generally leads to more Picard iterations for convergence. That being said, we think it may be possible that more steps can help reduce the number of Picard iterations, since more steps means that each step involves an easier prediction, which may translate to faster convergence.

---

> ### Comment · Reviewer_psnQ · 2023-08-11
>
> Thank you for the response, the extra experiments with 50-step DDIM look really promising and this has alleviated my concerns. I will keep my score as it is and continue to recommend acceptance.

---

### Official Review · Reviewer_RXY2 · 2023-07-06

**Soundness:** 2 fair
**Presentation:** 2 fair
**Contribution:** 2 fair
**Rating:** 6
**Confidence:** 3

**Summary:**

Instead of reducing the number of denoising steps, the paper proposes to  parallelize diffsuion denoising sampling via Picard iterations, by guessing the solution of future denoising steps and iteratively refining until convergence, which trades compute for speed. The authors then present ParaDiGMS to accelerate the sampling of pretrained diffusion models by denoising multiple steps in parallel and verify its effectiveness.

**Strengths:**

- The idea is interesting and may promote another way for accelerated sampling.
- The procedure of Algorithm is clear.

**Weaknesses:**

In total, I think there are still many problems for the paper. Please refer to the questions.

**Questions:**

**Q1**:  Some notations may be a bit confusing, since the $x_T$ is always used to represent the initial noise of diffusion and the authors use $x_0$ to represent the initial point.

**Q2**: For Algorithm 1, given the same denoising steps $T$, the proposed method is essentially similar to the existing techniques such as PNDM, DPMSolver. The existing high-order techniques should also cache the predicted score in previous time step, while the proposed method compute all the predicted score in a small window and use them for next selected window.

**Q3**: The number of model evaluations: For all the tables, do you choose a different $T$ and use more parallel iterations compared to the baseline to match the sampling performance? If so, what is the value of $T$ ? If I am not wrong, $T = Model Eval / Para Iter$ ? The details are not clear.

**Q4**: Following the prior question, how about setting a same $T$? Can parallel sampling improve the performance (e.g. FID)? This results could be important. If the performance remain the same, people can directly use a smaller denoising step $T$.

**Q5**:  For image generation,  does $batch size = 80$ mean the author uses a batch window size of $80$?

**Q6**: The evaluation results of image generation are not sufficient. How about the commonly-used FID for stable-diffusion? And more qualitative results should be shown and analysed.

**Q7**: The algorithm seems not practical. As the authors said, this method may consume more resources. However, **Q4** is not clear. If the performance remain the same, more resources should be used to maximize sample throughput instead of parallel computing.

**Q8**: Related works should discuss more parallelization techniques and the differences of between the proposed method and similar parallelization techniques.

**Limitations:**

The authors have adequately addressed the limitations.

---

> ### Author Rebuttal · Authors · 2023-08-09
>
> We thank the reviewer for the review. However, we believe that there is a misunderstanding as to the nature of our proposed method. ParaDiGMS is the first parallel sampling method for diffusion models: computing multiple denoising steps at the same time. This is fundamentally different from existing higher-order solvers, which uses multiple past timesteps to compute a single denoising step.
>
> It appears that Q2, Q3, Q4, Q7, Q8 stem from this misunderstanding. Please see the responses below.
>
> &nbsp;
>
> > **Q1: the authors use x_0 to represent the initial point**
>
> Yes, we make a note of this on L91 of the submission.
>
> &nbsp;
>
> > **Q2: For Algorithm 1, given the same denoising steps T, the proposed method is essentially similar to the existing techniques such as PNDM, DPMSolver**
>
> No, this is not the case. Our proposed method is fundamentally different from existing techniques such as PNDM, DPMSolver since our method denoises in parallel. In fact, since our method and existing techniques are addressing orthogonal issues, they can even be combined (e.g. ParaDiGMS + DPMSolver = ParaDPMSolver), as we demonstrate in our paper.
>
> To elaborate, these existing higher-order methods use multiple previous timesteps to predict a single denoising step. However, they are still sequential in nature, predicting one denoising step at a time. In contrast, our method predicts multiple steps in parallel, where each step can either be predicted using just the previous step (e.g. ParaDDIM) or using higher-order solvers (e.g. ParaDPMSolver).
>
> &nbsp;
>
> > **Q3: do you choose a different T and use more parallel iterations compared to the baseline to match the sampling performance? If so, what is the value of T? If I am not wrong, T = ModelEval/ParaIter? The details are not clear.**
>
> No, we use the same T when comparing sequential and parallel sampling. For example, for DDPM we use T=1000. The number of parallel iterations is lower than T because we are computing multiple steps in parallel.
>
> ModelEval/ParaIter does not equal T, but is rather (approximately) the batch window size, i.e., the number of steps we compute in parallel. The relationship is not exact because the batch window gets truncated near the endpoints of the trajectory.
>
> &nbsp;
>
> > **Q4: how about setting a same T? Can parallel sampling improve the performance (e.g. FID)?**
>
> We are already using the same T. We would not expect parallel sampling to improve performance since we are using the same T. The goal of our method is to improve sample latency.
>
> &nbsp;
>
> > **Q5: does batchsize=80 mean the author uses a batch window size of 80?**
>
> Yes, batchsize refers to “batch window size”. We will update this.
>
> &nbsp;
>
> > **Q6: How about the commonly-used FID for stable-diffusion? And more qualitative results should be shown and analysed**
>
> For stable-diffusion the commonly used metric is CLIP score since it is a text-to-image task. We show the CLIP score in Table 4 of our paper.
>
> For other tasks such as unconditional generation with LSUN, we show FID score in Table 5 of our paper.
>
> We show some qualitative results in Figure 4, where we see that the generated images are of high quality. We can include more image samples in the updated version of our paper.
>
> &nbsp;
>
> > **Q7: The algorithm seems not practical. As the authors said, this method may consume more resources. However, Q4 is not clear. If the performance remain the same, more resources should be used to maximize sample throughput instead of parallel computing.**
>
> It is unclear why the reviewer believes the algorithm is not practical. Our results show a very concrete 2-4x speedup in sampling latency across a range of diffusion policy and diffusion image generation tasks.
>
> The goal of our method is not to improve performance or sample throughput, but to improve sample latency. Our method requires more computation, but for many applications sample latency is critical and/or users are insensitive to the cost of compute. Please see the shared response on “Focus on latency” for more details.
>
> &nbsp;
>
> > **Q8: Related works should discuss more parallelization techniques and the differences of between the proposed method and similar parallelization techniques.**
>
> To our knowledge, our work is the first parallelization technique for diffusion model sampling, so there are no other similar techniques to compare with.
>
> &nbsp;
>
> We hope this addresses your concerns, and we are happy to answer any additional questions regarding our paper. Please consider updating your score in light of the initial review’s misunderstanding of our work.

---

> > ### Comment · Reviewer_RXY2 · 2023-08-15
> > **Thanks for the responses**
> >
> > Thanks for the responses. Most of questions have been addressed. And I believe that the idea is interesting and can bring new direction to the diffusion sampling community. However,
> > - Besides CLIP score, FID is also a common metric for text-to-image to evaluate the image quality. The evaluation is usually done in a zero-shot setting on text-image datasets of natural images such as COCO. The readers may concern about the image quality as well.
> > - When it coming to the balance between latency & performance, one would expect higher performance with lower latency.  As we known, reducing the number of sampling steps could be an obvious way to reduce the latency. Therefore, the comparison(e.g. FID、CLIP score) of different methods with the same latency is important, too.

---

> > > ### Author Response · Authors · 2023-08-15
> > > **Thanks for the responses**
> > >
> > > Thank you for the additional comments. We're glad to hear that most of the questions have been addressed.
> > >
> > > - **Text-to-image FID score:** Sure, we can evaluate zero-shot FID score on text-to-image on the COCO dataset. We took 5k images from the validation set, and drew 5k samples with 100-step DDIM.\
> > > \
> > > Here is the table from before with an extra column for FID score (lower is better). We note that though the FID score for our parallel sampling method is slightly better, we believe this is only due to natural variations in the evaluation metric. \
> > > \
> > > Stable Diffusion v2-0
> > > | Method          | time (s)         | CLIP (↑) | FID score (↓) 5k | Parallel Iters |
> > > |-----------------|--------------|------|------|----------------|
> > > | DDIM 100 steps  |              |      |                |
> > > | sequential      | 5.34         | 31.9 | 25.0 | 100            |
> > > | parallel w/ tolerance 5e-2  | 1.96 (**2.7x**)  | 31.9 | 24.4 | 19             |
> > >
> > > &nbsp;
> > >
> > > - **Latency & Performance:** Yes, we agree that when matching latency, our method should give better performance. We point to Table 5 in our paper, where it shows that 1000-step ParaDDPM gives both **better latency** (8.2s vs 12.2s) and **better performance** (12.9 FID vs 15.7 FID on 5k samples) when compared to sequential 500-step DDIM.
> > >
> > > &nbsp;
> > >
> > > We hope this addresses the additional comments from the reviewer. We're pleased that the reviewer believes the idea is interesting and can bring new direction to the diffusion sampling community, and hope the reviewer will consider updating their score if their concerns have been addressed.

---

> > > > ### Comment · Reviewer_RXY2 · 2023-08-16
> > > > **Thanks for responses**
> > > >
> > > > Thank the author for addressing my questions. My concerns about the performance have been resolved and I am willing to increase my score from 4 to 6.

---

### Official Review · Reviewer_veyN · 2023-07-06

**Soundness:** 4 excellent
**Presentation:** 4 excellent
**Contribution:** 3 good
**Rating:** 7
**Confidence:** 4

**Summary:**

The paper proposes an approach for speeding up sampling of diffusion models using Picard iterations. Multiple time steps in the sample process are predicted in parallel, iteratively refining until converging. Rather than refining all time steps at once which would not be practical, a sliding window approach is used: points in the window are refined then the window is moved once the later time steps have converged. Experimentally this approach is tested on diffusion policy learning and image generation benchmarks, where it is shown to provide substantial speedup over sequential sampling methods, while achieving comparable sample quality.

**Strengths:**

- Using Picard iterations to speed up diffusion model sampling makes sense and is useful in many low latency scenarios such as image editing.
- Showing that in the worst case, the approach will converge faster at least as fast as sequential sampling is nice (Lines 127-137). Similarly, providing tolerance bounds is useful (Lines 161-164).
- A good set of experiments/ablations on a variety of benchmarks are provided. The approach is clearly shown to be faster than sequential approaches, while providing similar quality samples.
- The approach is tested with multiple diffusion samplers, including the recent DPMSolver, showing that it can provide speedup even to already fast solvers.

**Weaknesses:**

- It is mentioned that a more relaxed tolerance value can be used when determining convergence (line 166). It would be useful to see the effect of that value on sampling times/image quality.
- Currently multiple GPUs are required to achieve net speedup on Stable Diffusion (Figure 4f).
- The approach is only useful for reducing latency; if generating lots of samples, then sampling batches sequentially is more efficient. However, as mentioned in the strengths, there are many applications where this is useful.

**Questions:**

- In Algorithm 1 on line 9, new points are initialised from the latest point in the window. Would it make more sense to initialise the values based on the prediction of $x_0$?
- What practically is the impact of using different tolerance values?

**Limitations:**

Limitations are well discussed in section 4.2.1 and section 5. It’s also worth noting that due to the extra compute, this approach will use more energy so has a negative environmental impact.

---

> ### Author Rebuttal · Authors · 2023-08-09
>
> We thank the reviewer for the review.
>
> &nbsp;
>
> > **It would be useful to see the effect of tolerance on sampling times/image quality. What practically is the impact of using different tolerance values?**
>
> Thanks for the suggestion. Here are additional experiments on a sweep over tolerance values for 200-step DDIM. We can see that the sample quality starts to degrade at a tolerance of 5e-1. We can use a tolerance of 1e-1 to maintain the same quality (or 5e-2 to be safe), and still achieve a sizable speedup compared to the sequential baseline.
>
> Stable Diffusion v2-0
> | tolerance | CLIP | time | Parallel Iters |
> |-----------|------|------|------------|
> | DDIM 200 steps|           |      |      |
> | sequential   | 31.9      | 10.3 | 200  |
> | 5e-3         | 31.9      | 7.4 (1.4x) | 36   |
> | 1e-2         | 31.9      | 5.1 (2.0x) | 28   |
> | 5e-2         | 31.9      | 4.9 (2.1x) | 21   |
> | 1e-1         | 31.9      | 3.8 (**2.7x**) | 16   |
> | 5e-1         | 31.5      | 2.1 (4.9x)  | 13   |
> | 1e-0         | 31.5      | 1.9 (5.4x) | 12   |
>
> &nbsp;
>
> > **Currently multiple GPUs are required to achieve net speedup on Stable Diffusion**
>
> Yes, multiple GPUs provide enough parallel computation to achieve net speedup on Stable Diffusion.
>
> &nbsp;
>
> > **only useful for reducing latency**
>
> Please see the shared response on “Focus on latency”. Our work indeed focuses on improving sample latency.
>
> &nbsp;
>
> > **New points are initialised from the latest point in the window. Would it make more sense to initialise the values based on the prediction of x_0?**
>
> This is a great suggestion! We tried some initial explorations on improving the initialization by extrapolating the trajectory using the prediction of x_0, but have not noticed consistent improvements over the current choice of copy initialization. The new suggestion is sometimes better but sometimes worse, so it may require more tinkering. In general, we do agree that the choice of initialization is a promising direction for further improvements, since the current choice of copy initialization is likely suboptimal.

---

> > ### Comment · Reviewer_veyN · 2023-08-14
> > **Response to Authors**
> >
> > Thanks to the authors for their responses; it is great to see the improvement in the low step scenario, this definitely strengthens the approach making it applicable in more scenarios. Additionally, I appreciate the evaluation of the tolerance value, I found this very informative. And I thank the authors for the comments on different initialisation strategies, it is interesting to hear that it is not always helpful. After these comments and the responses to the other reviewers, I am happy to increase my rating to accept.

---

### Official Review · Reviewer_PoCx · 2023-07-07

**Soundness:** 4 excellent
**Presentation:** 4 excellent
**Contribution:** 3 good
**Rating:** 7
**Confidence:** 4

**Summary:**

The authors propose a technique to reduce the time taken to sample from a diffusion model at the expense of using more FLOPs. Roughly speaking, the authors parallelize sampling by "guessing" xt at numerous values of t simultaneously, then computing the score function for each of these values of xt in parallel, and then repeating to refine the estimate of each xt over multiple steps.

**Strengths:**

- The paper is clearly written.
- The proposed method is novel.
- The proposed method is practically useful in situations where having low latency is important (e.g. iterative human-in-the-loop image generation).
- The experiments are thorough, showing speed-ups on a range of domains and even showing speedups when efficient samplers like DPMSolver are used.

**Weaknesses:**

Overall I am satisfied that this paper makes a tangible contribution. I have a couple of questions though about certain scenarios in which it is not immediately clear that this method is useful.
- I would be interested to see the performance of the latent image generation when less steps are used. E.g. for Stable diffusion (https://github.com/CompVis/stable-diffusion), the given example command "python scripts/txt2img.py --prompt "a photograph of an astronaut riding a horse" --plms " produces good images in only 50 steps, which is much less than the 200 used in Section 4.2.1. Does the proposed method still provide a gain in this setting?
- How does this method interact with other techniques to speed up sampling like progressive distillation (https://arxiv.org/abs/2202.00512)? I appreciate that one benefit of the proposed method is that it avoids the need to have any sort of "distillation" training phase, but can the proposed method provide additional advantages when used in combination with a "progressively distilled" model?

**Questions:**

See above.

**Limitations:**

The limitations are adequately addressed.

---

> ### Author Rebuttal · Authors · 2023-08-09
>
> We thank the reviewer for the review.
>
> &nbsp;
>
> > **less steps are used.**
>
> Please see the shared response on “Speedup with few steps”. We present exciting developments where we implement custom multiprocessing for ParaDiGMS and are now able to see a 2.5x speedup on the most widely used setting of 50-step DDIM, and a 1.3x speedup even for 25-step DDIM.
>
> &nbsp;
>
> > **How does this method interact with progressive distillation**
>
> In terms of compatibility, our method can be used in combination with progressive distillation. The distilled models are sampled using sequential denoising, which can similarly be parallelized using ParaDiGMS. Since the distilled models use even fewer steps (e.g. 4 or 8), achieving speedups with parallel sampling may be more challenging, but we believe it may be possible with future improvements.
>
> In terms of differences, as you correctly point out progressive distillation requires retraining, whereas our method is an off-the-shelf sampling method that does not require any additional training. It is also important to note that distillation often leads to slightly worse sample quality, whereas our method is able to maintain the same sample quality.

---

> > ### Comment · Reviewer_PoCx · 2023-08-12
> >
> > Thank you for the response - it addressed my comments and I will continue to recommend that this paper is accepted.

---

### Official Review · Reviewer_gnmk · 2023-07-08

**Soundness:** 3 good
**Presentation:** 3 good
**Contribution:** 3 good
**Rating:** 6
**Confidence:** 4

**Summary:**

Naive parallelization can let us generate multiple samples which improves throughput. However, the wall-clock time remains the same. To reduce the wall-clock time during diffusion model sampling, this paper proposes ParaDiGMS, a parallel sampling method of diffusion models based on Picard Iteration which improves latency. ParaDiGMS is compatible with classifier-free guidance and prior fast sampling methods. The experiment results show that ParaDiGMS can achieve a speedup without quality degradation.

**Strengths:**

1. The authors provide a theoretical guarantee and error analysis of the proposed method.

2. The paper is motivated by the classical Picard Iteration which brings new ideas to the diffusion sampling community.

3. The paper is easy to follow and overall well-written which helps the reviewer understand the content.

4. The experiment results show that it achieves a speedup with no measurable decrease in quality.

**Weaknesses:**

The applicability of the proposed method is limited to certain situations. The total evaluations of ParaDiGMS are approximately twice that of the baseline method according to the experiments. Therefore, the proposed method may not be suitable for situations where maximizing sample throughput is a priority.

**Questions:**

In the section on experiments of diffusion image generation, the authors test ParaDiGMS on the settings of 1000 NFEs for DDPM, 200/500 NFEs for DDIM, and 200 NFEs for DPMSolver. It is questionable whether ParaDiGMS can achieve a similar speedup in fewer NFEs(like around 50 NFEs) since fewer NFEs are commonly used in downstream applications and the state-of-the-art samplers can achieve comparable results in this setting.

**Limitations:**

The author addressed the limitations of their work.

---

> ### Author Rebuttal · Authors · 2023-08-09
>
> We thank the reviewer for the review.
>
> &nbsp;
>
> >  **may not be suitable for situations where maximizing sample throughput is a priority**
>
> Please see the shared response on “Focus on latency”. Our work indeed focuses on improving sample latency.
>
> &nbsp;
>
> > **It is questionable whether ParaDiGMS can achieve a similar speedup in fewer NFEs**
>
> Please see the shared response on “Speedup with few steps”. We present exciting developments where we implement custom multiprocessing for ParaDiGMS and are now able to see a 2.5x speedup on the most widely used setting of 50-step DDIM, and a 1.3x speedup even for 25-step DDIM.

---

> > ### Comment · Reviewer_gnmk · 2023-08-18
> >
> > Thank you for your detailed response. The acceleration on 50 steps DDIM is exciting. However, the performance on Stable Diffusion evaluated by FID/Clip saturates fast. As can be seen in the table, 25 steps DDIM achieve a Clip score of 31.7, which is close to the performance of 50 steps DDIM, 31.9. Could the author provide quantitative results on LSUN church (as in your paper) to better show the speed-up of ParaDiGMS?

---

> > > ### Author Response · Authors · 2023-08-21
> > > **Comment**
> > >
> > > Thanks for the suggestion! Here are quantitative results on LSUN church for 100/50/25 steps. We similarly compute FID score using 5k samples.
> > >
> > > \
> > > ddpm-ema-church-256
> > > | Method          | time (s)         | FID score (↓) 5k | Parallel Iters |
> > > |-----------------|--------------|------|----------------|
> > > | DDIM 100 steps  |              |                 |
> > > | sequential      | 2.48         |  15.10 | 100            |
> > > | parallel w/ tolerance 1e-3  | 0.99 (**2.5x**)  |  14.78 | 23             |
> > > | DDIM 50 steps  |              |                 |
> > > | sequential      | 1.24         |  15.33 | 50            |
> > > | parallel w/ tolerance 1e-3  | 0.73 (**1.7x**)  |  15.68 | 17             |
> > > | DDIM 25 steps  |              |                 |
> > > | sequential      | 0.61         |  15.59 | 25            |
> > > | parallel w/ tolerance 1e-3  | 0.55 (**1.1x**)  |  15.86 | 15             |
> > >
> > > &nbsp;
> > >
> > > We can see that indeed pixel space diffusion is more challenging than latent space diffusion, and we noticed that using a lower tolerance (1e-3) was more suitable for pixel space diffusion. In terms of FID score, our method matches the performance for 100 steps (actually is slightly better in this measurement), and is slightly worse for 50 and 25 steps. In terms of speed, for 100-step DDIM we see a 2.5x speedup, for 50-step DDIM we see a 1.7x speedup, and finally for 25-step DDIM we see a 1.1x speedup. This matches the trends from latent diffusion experiments, where we also saw the speedups drop at around 25 steps.
> > >
> > > To summarize these results, pixel-space diffusion is less forgiving in terms of using few-step sampling. Fortunately, 100-step ParaDDIM can provide a significant (2.5x) boost in speed while maintaining higher sample quality than 50 or 25 step DDIM.

---

### Author Rebuttal · Authors · 2023-08-09

Thank you all for the helpful reviews. We are happy to hear that our contribution “brings new ideas to the diffusion sampling community”, is “practically useful”, and is “highly significant and…will be widely used”. We also appreciate that reviewers found the writing “easy to follow and overall well-written”, with a “good set of experiments/ablations on a variety of benchmarks”.

&nbsp;

First, we have some exciting developments regarding speedups for low-step sampling. **We optimized our implementation for multi-GPU inference, and are now able to sample 2.5x faster on 50-step DDIM on stable diffusion (details below)!** This is very exciting as 50-step DDIM is the most popular setting used for stable diffusion.

&nbsp;

Next, we discuss two common points raised by the reviewers.

1. **Speedup with few steps** \
\
First, we note that our method shows around 2x speedup even when using very few steps (15-step DDIM and 15-step DPMSolver) on the diffusion policy experiments. For diffusion policy, the models are smaller so a single GPU can handle parallel computation without much slowdown. \
\
Since stable diffusion is a larger model, the FLOPS saturation of the GPU plays a large role, so running with multiGPU is necessary. Our initial implementation using torch.DataParallel had a large overhead when using fewer steps, so we were unable to see any speedups for 50-step DDIM. Torch DistributedDataParallel also did not work since our use case is inference and not training. \
\
Recently, we implemented custom multiprocessing inference for ParaDiGMS, and we are able to see a substantial speedup (2.5x) for DDIM 50 steps! This is very exciting since, as the reviewers noted, 50-step inference is the most popular and the default setting for stable diffusion. \
\
As suggested by the reviewers, we run additional experiments to evaluate our algorithm on 100-step, 50-step, and 25-step DDIM for stable diffusion. \
\
We note that when using fewer steps, the tolerance level should be lowered to maintain the same sample quality. For 100-step DDIM, tolerance 5e-2 achieves the same sample quality with 2.7x speedup. For 50-step DDIM, tolerance 5e-2 achieves the same sample quality with 2.5x speedup. For 25-step DDIM, tolerance 1e-2 achieves the same sample quality with 1.3x speedup. \
\
Stable Diffusion v2-0
| Method          | time         | CLIP | Parallel Iters |
|-----------------|--------------|------|----------------|
| DDIM 100 steps  |              |      |                |
| sequential      | 5.34         | 31.9 | 100            |
| tolerance 5e-2  | 1.96 (**2.7x**)  | 31.9 | 19             |
| tolerance 1e-1  | 1.68 (3.1x)  | 31.6 | 17             |
| DDIM 50 steps   |              |      |                |
| sequential      | 2.62         | 31.9 | 50             |
| tolerance 5e-2  | 1.05 (**2.5x**)  | 31.9 | 17             |
| tolerance 1e-1  | 0.93 (2.8x)  | 31.3 | 15             |
| DDIM 25 steps   |              |      |                |
| sequential      | 1.31         | 31.7 | 25             |
| tolerance 1e-2  | 0.99 (**1.3x**)  | 31.7 | 17             |
| tolerance 5e-2  | 0.76 (1.7x)  | 31.4 | 13             |

 2. **Focus on latency**\
\
As stated in the paper, and noted by many reviewers, the goal of our method is to improve sample latency but not throughput. For many applications such as interactive generation or real-time policy execution, sample latency is much more critical. Moreover, many users are insensitive to the cost of compute during inference, since the GPU demands during inference are much lower than that during training. Finally, we note that the reviewers are in agreement with us that “there are many applications where [improving latency] is useful”.


&nbsp;

We are extremely excited about the immediate benefits of ParaDiGMS – a 2.5x speedup on the default setting of stable diffusion. We are even more excited about the general potential of parallel sampling and the new avenue of research it unlocks for diffusion models. We fully agree with the reviewers on the significance of the idea that our paper introduces to the community, and are eager for future improvements on aspects such as discretization, batch window initialization, or multiprocessing optimization.

---

### Decision · Program_Chairs · 2023-09-21

**Decision:**

Accept (spotlight)

**Comment:**

The reviewers' assessment of the paper is exceptionally positive. They commend the practicality and significance of using Picard iterations for swift diffusion model sampling, particularly in time-sensitive scenarios like image editing. The paper's strength lies in demonstrating both faster convergence and tolerance bounds, supported by a thorough experimentation process that establishes its efficiency and comparable sample quality. The reviewer acknowledges the approach's superiority over sequential methods, even for already efficient solvers like DPMSolver.